# Behavioural and Cognitive Attitudes of Paediatricians towards Influenza Self-Vaccination—Partial Mediation Model

**DOI:** 10.3390/vaccines10081206

**Published:** 2022-07-29

**Authors:** Tomasz Sobierajski, Helena Bulińska-Stangrecka, Monika Wanke-Rytt, Paweł Stefanoff, Ewa Augustynowicz

**Affiliations:** 1Faculty of Applied Social Sciences and Resocialization, University of Warsaw, Krakowskie Przedmieście 26/28, 00-927 Warsaw, Poland; 2Faculty of Administration and Social Sciences, Warsaw University of Technology, Plac Politechniki Street 1, 00-661 Warsaw, Poland; tomasz.sobierajski@wp.pl; 3Department of Pediatrics with Clinical Assessment Unit, Medical University of Warsaw, Żwirki i Wigury 61, 02-091 Warsaw, Poland; monika.wanke@uckwum.pl; 4Division of Infection Control and Environmental Health, Norwegian Institute of Public Health, Lovisenberggata 8, 0456 Oslo, Norway; pawel.stefanoff@gmail.com; 5Department of Epidemiology of Infectious Diseases and Surveillance, National Institute of Public Health-National Institute of Hygiene, Chocimska 24, 00-791 Warsaw, Poland; eaugustynowicz@pzh.gov.pl

**Keywords:** self-vaccination, influenza, attitudes, behaviour, cognitive, paediatricians

## Abstract

(1) Background: This study aims to analyse the attitudinal components influencing paediatricians’ self-vaccination. (2) Methods: The national-cross survey was conducted among paediatricians involved in childhood vaccination within the immunisation program. (3) Results: A hypothetical model indicating the influence of cognitive and behavioural factors on influenza vaccination among paediatricians was verified based on a survey of Polish paediatricians. A simple mediation model, based on Triandis’ Theory of Interpersonal Behaviour, reflects a relationship in which knowledge and beliefs about outcomes contribute to whether paediatricians vaccinate against influenza. (4) Conclusions: The presented research shows that the pro-vaccination behaviours of paediatricians are not only influenced by cognitive factors but also the behavioural components of attitudes, which are equally important. The conclusions point to the pivotal role of shaping both knowledge and understanding of the effectiveness of immunisation programmes in building the pro-vaccination attitudes of paediatricians. (5) Practical Implications: This is the first representative study of Polish paediatricians to demonstrate how their attitudes and behaviour are related to self-vaccination. Its conclusions allow policymakers to develop programmes to support effective measures against the spread of infectious diseases through the self-vaccination of medical professionals.

## 1. Introduction

Immunisation is a safe way to prevent harmful infections and fight against infectious diseases [1]. The association between paediatricians’ positive attitudes towards vaccination and self-vaccination can result in several ways, including more significant support for vaccination among children [2] or more effective immunization recommendations [3]. Extant studies suggest that healthcare providers play a crucial role in overcoming vaccine hesitancy [4]. Furthermore, it has been indicated that paediatricians’ accurate vaccine-related knowledge can effectively address parental vaccine hesitancy [5]. Healthcare providers affect parents’ attitudes toward vaccination, but they are also treated as a trusted source of knowledge about vaccination and influence their behaviours [6].

Moreover, various studies imply that parents’ decisions regarding vaccinations depend on healthcare provider recommendations [7,8]. Hence, understanding paediatricians’ attitudes toward vaccination are essential for shaping processes promoting immunisations. This may indicate a need to better comprehend the factors that increase the acceptance of vaccination among paediatricians.

Increasing acceptance of immunisation programmes and physicians’ self-vaccination for influenza is paramount in Poland, where past data indicate low vaccination rates among physicians. According to current research, only 16–20% of medical personnel are vaccinated against influenza [9,10]. For this reason, it is essential to understand the mechanisms that boost positive attitudes toward vaccination among physicians. Paediatricians’ support for vaccinations in Poland was associated with increasing age, exclusive use of scientific sources of information, and addressing vaccination myths [11]. Furthermore, recent studies show that only a small percentage of doctors recommend all available vaccinations [12]. Therefore, it is crucial to understand how to effectively influence the immunisation attitudes of healthcare providers. Additionally, there is a moral obligation to self-vaccinate against influenza and other infectious diseases by doctors. This applies to disease prevention and public health ethics and, above all, to modelling positive behaviour among patients [13,14]. Therefore, self-vaccination among physicians is a significant factor influencing patients’ behaviours and situations in health care on a macro scale.

We aimed to extend previous studies and analyse the underlying mechanism supporting paediatricians’ positive attitude towards vaccination. Besides, we intend to comprehend the relationship between factors leading to positive cognitive and behavioural attitudes towards influenza vaccination among Polish paediatricians. In our study, we examine the role of attitudes in behaviour formation. We investigate paediatricians’ attitudes toward vaccination and their behavioural outcomes regarding influenza immunization.

This study aims to answer how all the components of paediatricians’ attitudes towards influenza self-vaccination are shaped in Poland. This objective contributes to the formation of appropriate interventions to prevent the problem of low influenza self-vaccination among physicians.

## 2. Assumptions of Theory of Interpersonal Behaviour

The Materials and Methods should be described with sufficient detail to allow others to consider various factors that influence doctors’ attitudes to vaccination, contributing to the labile position of some healthcare providers [15,16]. Therefore, understanding the mechanisms supporting the positive attitudes of paediatricians towards vaccination is a crucial challenge for contemporary medical policymakers.

Attitudes are characterised as cognitive generalisations that lead to individuals’ responses to the situational context [17]. It is an idea of “predisposing actions” [18]. Based on Triandis’ Theory of Interpersonal Behaviour (TIB), we seek to explain the factors affecting paediatricians’ outlook toward influenza vaccinations. This theory links individual behaviour to various antecedents such as attitudes, social factors, affective elements, and habits [17]. In this approach, attitudes are perceived as a dynamic process connecting an object with an anticipated outcome and placing value on this outcome. In other words, people are more likely to engage in a particular behaviour if they perceive it as beneficial [18].

According to TIB [18], attitudes comprise two elements: beliefs about outcomes and evaluation of outcomes. The first component refers to cognitive perception and deeply held convictions. It encompasses individual beliefs and knowledge about an attitude’s object. It manifests itself in the way an individual understands a given phenomenon. Regarding vaccination issues, it means how a paediatrician perceives their attitude towards immunizations. The second component evaluates an outcome, which means it uses the behavioural approach in conceptualising an individual’s predispositions toward specific issues. This includes how a person assumes it is expected or predicted behaviour in specific situations. The behavioural attitude towards vaccination is manifested in professional activities, such as recommending vaccination to patients. Due to the nature of the study, we have adopted a perspective that aims to verify whether a cognitive component (beliefs about the outcome) of paediatricians’ outlook is associated with their flu vaccination. We are going to explain if this relationship is in some way mediated by the behavioural aspect of attitudes (evaluation of outcome). Consequently, we intend to discuss the mechanism indicating the complex relationship that underlies the self-vaccination of doctors. In line with this theory, we examine how the relationship between paediatricians’ declared cognitive and behavioural attitudes affects their behaviours regarding influenza immunizations.

### Hypotheses

Based on the analysis of the material presented in the Introduction and assumptions of the Theory of Interpersonal Behaviour, we adopted three research hypotheses:

**H1.** *Cognitive attitude positively affects influenza self-vaccination among physicians*;

**H2.** *Behavioural attitude positively affects influenza self-vaccination among physicians*;

**H3.** *Behavioural attitude mediates the link between cognitive attitude and self-vaccination among physicians*.

## 3. Materials and Methods

### 3.1. Design and Settings

We obtained data from a national-cross survey of Polish paediatricians. We conducted a national survey of paediatricians involved in vaccinating children as outlined in the Polish National Vaccination Programme (NVP) [19]. Under the NVP, vaccinations are divided into: mandatory vaccinations for children and adolescents (e.g., against tuberculosis with intradermal BCG vaccine, against diphtheria, tetanus, and whooping cough, against poliomyelitis or measles, mumps, and rubella), vaccinations for children and adults who are particularly exposed to infection due to clinical or epidemiological reasons (e.g., against chickenpox), and vaccinations for children and adolescents who are particularly exposed to infection due to clinical or epidemiological reasons (e.g., against cowpox) [19]. Mandatory vaccinations are required by law, and these vaccinations are financed by the state treasury. Recommended vaccinations are voluntary and are not reimbursed by the state. The survey was conducted between June and July 2017.

### 3.2. Participants and Sample Size

The main emphasis in the random selection of respondents was on paediatricians who qualify for vaccination and vaccinate children. As a result, 500 doctors meeting the criteria were selected for the appropriate examinations, 97 of whom were family doctors but did not have specialisation in paediatrics. Therefore, for further analysis, we took only a group of 403 doctors who met two conditions: they qualified children for vaccination and vaccinated children, and they had a specialisation in paediatrics.

A simple random selection of doctors for our study was challenging because we had no access to a list of doctors who were paediatricians and vaccinated children. According to the data of the Supreme Chamber of Physicians and Dentists, the paediatrician specialisation has about 17,000 people, of whom about 15,000 are paediatricians. The selection of doctors for the examination was random and multistage. In each province, the number of paediatricians was selected to reflect the number of inhabitants. Then the interviewers drew lots of medical institutions that provided paediatric care. In the selected institutions, they contacted the paediatrician and conducted the examination properly. The examined person and the head of the institution where the paediatrician works had to agree to conduct the examination. In addition, the prerequisite for the examination was that the paediatrician qualified children for vaccination and vaccinated them.

### 3.3. Data Collection

The survey was based on a computer-assisted telephone interview (CATI). CATI was based on a previously prepared structured questionnaire. The DSC Research Group conducted the survey. The study was preceded by a pilot study in which ten randomly selected paediatricians participated. The pilot study results, and the methodological guidelines were communicated to the research company and the experienced interviewers who conducted the study.

### 3.4. Research Instrument

The questionnaire consisted of 17 questions. The first four had a sociodemographic nature (gender, age, and seniority), and the remaining questions concerned, among other things, self-assessment of knowledge about vaccination, sources of information about vaccination, participation in vaccination training, informing the child’s parents about recommended vaccinations, problems encountered by doctors in their daily practice when vaccinating children, and attitudes to common social myths about vaccination. Almost all the questions were closed questions, with possible answers or points on the scale. Only one question was pre-categorised. Some questions used a numerical scale, and some answers were dichotomous or alternative.

In the questionnaire used in this research, the question about the cognitive attitude representing knowledge based on deeply held convictions and beliefs has been formulated in the form of a question about the self-determination of one’s attitude towards vaccination, where the respondents could answer on a five-stage scale: “I am a strong supporter of vaccination”, “I am a moderate supporter of vaccination”, “I am a moderate opponent of vaccination”, “I am a strong opponent of vaccination”, and a neutral one: “I am neither a supporter nor an opponent of vaccination”.

The question relating to the behavioural attitude concerned the determination of the potential behaviour of a paediatrician regarding the vaccination of their newborn child. It contained four possible answers: “I only carry out mandatory vaccinations”, “I carry out mandatory and selected recommended vaccinations”, “I carry out mandatory and all recommended vaccinations”, and “I will not vaccinate my child at all”.

The question about the self-vaccination of paediatricians was formulated as follows: “Do you vaccinate against flu?” and respondents could choose among three options: “I do, regularly every year or almost every year”, “I do, irregularly, from time to time”, or “I do not, I never get vaccinated”.

### 3.5. Statistical Analysis

The sample size was calculated, assuming the margin of error to be no more than 5% at the 95% confidence level. There are 17,000 paediatricians registered in Poland, and therefore, the minimum required sample is 376. Hence, the used sample of 403 responses meets the required criteria. Statistical analyses were carried out using RStudio, and the bivariate Pearson correlation coefficients were estimated to analyse construct validity.

### 3.6. Ethical Considerations

DSC Research Group is a research company that implemented and applied an information security management system compliant with the requirements of the ISO 27001 certificate, has a CATI studio compliant with the ISO 27001 certificate, and combines the procedures of the ISO 27001 information security management system with the requirements of the General Data Protection Regulation in UE. Study participants were informed that the study was anonymous and confidential, and no personal information, including computer IP, was collected. To avoid identifying the study participant, the results were collected and analysed collectively.

## 4. Results

### 4.1. Characteristics of Study Participants

A total of 403 paediatricians completed the survey, of which 80.9% (N = 326) were women, and 19.1% (N = 77) were men. The average age of respondents was 55 years old (this is consistent with the data on the average age of paediatricians in the Polish population and corresponds to the gender structure). Seniority in the profession of most of the respondents (89.1%, N = 359) was higher than 15 years. One in two paediatricians (52.4%, N = 211) rate their knowledge of immunisations as good, and four in ten (41.7%, N = 168) rate their knowledge of immunisations as very good (Table 1).

### 4.2. Cognitive and Behavioural Attitudes and Influenza Self-Vaccination

Most paediatricians surveyed (86.4%, N = 348) reported being strong supporters of vaccination, and only one person reported being neither a supporter nor an opponent of vaccination. The remaining respondents (13.4%, N = 54) reported being moderate supporters of vaccination. None of the doctors marked the answer “I am a moderate opponent of vaccination”.

There was a statistical relationship between the cognitive attitude toward vaccination and the age of the respondents (*p* < 0.001) (Table 2).

In the screening question, “Suppose you have to decide to vaccinate your newborn child with no contraindications to vaccination, what do you do?” only one person declared that they would not vaccinate their child.

Nearly one in two respondents (48.6%, N = 196) said they would do the complete vaccination package (mandatory and recommended). Almost the same number of people (48.1%, N = 194) declared that they would vaccinate their children with all mandatory and selected recommended vaccinations. Moreover, only 3% (N = 12) would vaccinate their children with mandatory vaccinations only. Women and paediatricians with a length of service between 10 and 15 years were more likely to vaccinate their children with the complete vaccination package (Table 3).

Influenza vaccination rates among Polish paediatricians are very high compared to influenza vaccination rates for the general adult Polish population (only 4.1% of all adult Poles were vaccinated against influenza in the 2019/2020 season) [10]. One in three surveyed paediatricians (62.5, N = 252) vaccinates against influenza regularly every year or almost every year. One in eight paediatricians (13.2%, N = 53) vaccinates irregularly from time to time. Moreover, one in four paediatricians (24.3%, N = 98) does not vaccinate against flu. Those with less than 10 years of professional experience are more likely not to vaccinate against flu (Table 4).

Among the responses to the deepening question, “For what reason do you not get vaccinated against influenza?” four types of responses predominated: 1. belief in natural solid immunity, for example—“I never get sick”; 2. belief in self-immunity, for example—“I acquire natural immunity from patients”; 3. doubt in flu vaccines, for example—“I am not convinced of the effectiveness of these vaccines”; and 4. experiences after vaccination, for example—“I got sick after being vaccinated and have not been vaccinated since”.

## 5. Discussion

Our study was designed to explore the relationship between attitudes and influenza self-vaccination among paediatricians. It has been found that both cognitive and behavioural attitudes facilitate influenza self-vaccination. This study suggests that to increase influenza self-vaccination among paediatricians, it is necessary to positively influence their knowledge in this area and their conviction of the desired results.

This research was conducted based on TIB [18], according to which two critical elements of attitudes are: a cognitive component and a behavioural component. The cognitive component includes beliefs and knowledge about outcomes. At the same time, the behavioural component contains an individual’s predisposition to act based on the evaluation of outcomes. Thus, according to the TIB [18], the tendency of an individual to a given behaviour of self-vaccination against influenza, depends on the conviction of the consequences of this behaviour as well as the knowledge and beliefs associated with this behaviour [20]. These studies have incorporated TIB into modelling causal relationships associated with self-vaccination by paediatricians, anticipating that cognitive and behavioural components influence doctors’ behaviour regarding flu vaccinations.

The analysis showed that none of the correlations were higher than the recommended level of r = 0.50 (the highest value was r = 0.371), indicating that it is unlikely to influence the results of further regression-based analyses [21]. Moreover, there were significant correlations between the variables, which suggests interrelations between the constructs (Table 5).

The next step was to analyse the mediation based on Hayes’ regression-based method and use the PROCESS procedure developed by Hayes [22]. The bootstrapping approach with 5000 bootstrap confidence intervals was used. According to this procedure, the relationship between cognitive attitude (X), behavioural attitude (M), and declared influenza self-vaccination of paediatricians (Y) was analysed. To test the significance of the mediation effect, the bootstrapping 95 percent confidence interval (CI) was used [23].

The results found that the total effect of cognitive attitude on actual self-vaccination of paediatricians (β = 0.61, *p* < 0.001) decreased when the mediator (behavioural attitude) was included in the model (β = 0.43, *p* < 0.001). This hypothesis was supported. The carried-out analysis confirmed the relationship between cognitive attitude and declared influenza self-vaccination of paediatricians mediated by behavioural attitude (Table 6).

A bootstrap estimation procedure (with a bootstrap sample of 5000) was used. As zero is not in the CI, the result is significant [23]. The relative magnitude of mediation effect (ratio of indirect to total effect of X on Y) was significant (β = 0.302, LLCI = 0.179, ULCI = 0.526).

The results provide evidence supporting H1 (*Cognitive attitude positively affects influenza self-vaccination among paediatricians*), H2 (*Behavioural attitude positively affects influenza self-vaccination among physicians*), and H3 (*Behavioural attitude mediates the link between cognitive attitude and self-vaccination among paediatricians*). This indicates partial mediation (Figure 1.)

The cognitive component of an attitude refers to knowledge and beliefs. Thus, in the context of immunisation, it concerns individual beliefs about the effectiveness of vaccines. Studies on paediatricians/physicians indicate that knowledge about vaccination positively affects their preference for self-vaccination against influenza [24]. The studies confirm that it is necessary to recommend influenza vaccination more strongly to patients and support their knowledge about the positive aspects of vaccination [25]. Therefore, the data above suggests that knowledge about vaccination increases its acceptance among medical practitioners. Therefore, our study has shown that a cognitive attitude positively influences the inclination to self-vaccination of paediatricians (H1).

The research indicates the relationship between behaviour and preferences for self-vaccination of medical personnel [26]. Physicians’ declaration of willingness or action to offer vaccinations is associated with their greater propensity to undergo vaccinations themselves, and it also positively affects patients’ perception of vaccination. Moreover, the conducted analyses suggest that the lack of conviction about vaccination effectiveness is a significant barrier among health care workers to accepting immunisation [27,28]. Moreover, there were acknowledged associations between recommendations and convictions about vaccination effectiveness with declared readiness for self-immunity against seasonal diseases [29]. The conviction of a positive vaccination outcome is one of the most important factors in increasing self-vaccination against influenza among healthcare professionals [30].

In summary, an individual’s conviction of the effectiveness of immunisation is an essential predictor of self-vaccination [31]. Also, the conviction of physicians about the positive effects of vaccination is the most significant predictor of vaccination recommendations [32]. Therefore, the conviction of the desired outcome can contribute to an increase in self-vaccination. In addition, our study found that behavioural attitude positively affects influenza self-vaccination among physicians (H2).

Although knowledge is considered an integral aspect of a positive attitude towards vaccination, few empirical studies analyse its links with self-vaccination [33]. However, paediatricians’ attitudes play a crucial role in predicting their willingness to self-immunise [14]. The knowledge and conviction about immunisation-efficacy positively influence physicians’ attitudes to vaccination, and this study assumes that both attitudinal components affect the propensity to self-vaccination. Moreover, it is assumed that knowledge and the assumption of an effective outcome determine the attitude of paediatricians to self-vaccination. And our study confirmed that behavioural attitude mediates the link between cognitive attitude and self-vaccination among physicians (H3).

Our findings support that paediatricians’ attitudes towards influenza self-vaccination are influenced by cognitive and behavioural components of attitudes.

Research indicates the need to change attitudes in the behavioural dimension [14]. However, our research indicates that in controlling a pandemic situation and reducing the incidence of infectious diseases, both behavioural and cognitive levels of attitude, that is, a belief in the effectiveness of vaccination and evaluation of outcomes, are crucial.

Though many researchers focus on strengthening healthcare providers’ knowledge to improve self-vaccination levels [34,35], this study suggests that increased self-vaccination among paediatricians can also be achieved by strengthening the behavioural component of the paediatricians’ attitude.

Education plays a vital role in this process [36]. However, to be effective, training needs to change at the cognitive level and concern the behavioural one by providing model behaviours that influence the belief in vaccination efficacy. This study confirms that it is necessary to organize trainings for doctors to better understand the positive outcomes of vaccination.

Therefore, a positive attitude at cognitive and behavioural levels plays a crucial role in increasing doctors’ inoculations. Shaping attitudes should be aimed at changing doctors’ knowledge regarding vaccination and influencing the internal belief in their effectiveness and legitimacy and, consequently, their behavioural attitude.

### Limitations of the Study

This study has two specific limitations. Each of them is a guideline for further research on the self-vaccination of paediatricians. The first limitation is that these are cross-sectional studies. Therefore, further studies may focus on a long-term analysis of the phenomenon, which will allow us to consider the real impact of various factors on paediatricians’ influenza vaccination behaviour. Another limitation is that the data was collected quantitatively, which allowed us to capture the effect of the phenomenon’s scale. However, examining paediatricians’ deepened attitudes regarding vaccination in the future is worthwhile. Therefore, further qualitative analysis will enable a deeper in-depth understanding of the paediatricians’ determinants of self-vaccination behaviour.

## 6. Conclusions and Practical Implications

The model presented here explains how vaccination acceptance and willingness can be increased by influencing specific components of paediatricians’ attitudes. This model can be used to analyse physicians’—not just only paediatricians’—attitudes toward vaccination and, as a result, patient persuasion toward vaccination. This is important, especially in the case of paediatric vaccinations and the increasing number of parental refusals to vaccinate their children in developed countries.

The cognitive change must ensure that physicians or paediatricians are convinced to vaccinate. Our model showing the role of simultaneously reinforcing paediatricians’ behavioural and cognitive attitudes toward vaccination plays a significant role when it is so important for the entire world to vaccinate children against COVID-19 and in the context of future potential pandemics that, like SARS-CoV-2, will require population-wide vaccination interventions. Doctors, including paediatricians, who work directly with children and indirectly with parents, have a crucial role in education about the need for SARS-CoV-2 vaccination. However, as our analysis shows, this is only possible if they receive adequate support for their attitudes towards vaccination. COVID-19 vaccination acceptance varies greatly around the world, ranging from 93.3% in Indonesia [37] to 80% in Denmark, 74% in Italy, 70% in Germany, 60% in France [38], 67% in the United States [39], and 31% in Poland [40].

The above model can improve the quality of vaccinology training for physicians and can be implemented to train paediatricians in different countries/cultural contexts.

## Figures and Tables

**Figure 1 vaccines-10-01206-f001:**
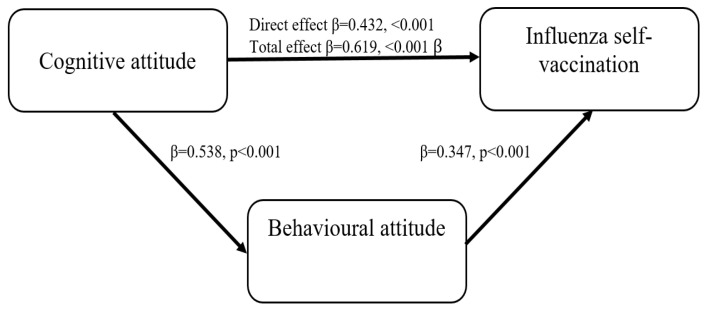
Illustration of the Partial Mediation Model of Paediatricians’ Influenza Self-Vaccination.

**Table 1 vaccines-10-01206-t001:** Characteristics of respondents (N = 403).

	N	%
Gender		
Female	326	80.9
Male	77	19.1
Age		
<40	39	9.6
41–50	66	16.4
51–60	158	39.2
61<	107	26.6
I do not want to specify	33	8.2
Seniority in the profession		
Up to 5 years	2	0.5
More than 5 to 10 years	23	5.7
More than 10 to 15 years	19	4.7
Over 15 years	359	89.1
Self-assessment of knowledge about vaccines		
Very bad	0	0.0
Bad	2	0.5
Average	18	4.5
Good	211	52.4
Very good	168	41.7
I do not know/hard to say	4	1.0

**Table 2 vaccines-10-01206-t002:** Approach to vaccination vs. demographic categories (N = 403).

	Strong Supporter of Vaccination	Moderate Supporter of Vaccination	Neither a Supporter Nor an Opponent of Vaccination	*p* Value
	N (%)			
Gender				
Female	287 (88.0)	39 (12.0)	0 (0.0)	0.024
Male	61 (79.2)	15 (19.5)	1 (1.3)	
Age				
<40	32 (82.1)	7 (17.9)	0 (0.0)	*p* < 0.001
41–50	52 (78.8)	14 (21.2)	0 (0.0)	
51–60	144 (91.1)	14 (8.9)	0 (0.0)	
61<	91 (85.)	16 (15.0)	1 (0.9)	
I do not want to specify	30 (90.9)	3 (9.1)	0 (0.0)	
Seniority in the profession				
Up to 5 years	1 (50.0)	1 (50.0)	0 (0.0)	*p* = 0.863
More than 5 to 10 years	20 (87.0)	3 (13.0)	0 (0.0)	
More than 10 to 15 years	16 (84.2)	3 (15.8)	0 (0.0)	
Over 15 years	311 (86.6)	47 (13.1)	1 (0.3)	
Self-assessment of knowledge about vaccines				
Very bad	0 (0.0)	0 (0.0)	0 (0.0)	*p* = 0.491
Bad	2 (100.0)	0 (0.0)	0 (0.0)	
Average	15 (83.3)	3 (16.7)	0 (0.0)	
Good	174 (82.5)	36 (17.1)	1 (0.5)	
Very good	153 (91.1)	15 (8.9)	0 (0.0)	
I do not know/hard to say	4 (100.0)	0 (0.0)	0 (0.0)	

Note. The distribution for the response “moderate opponent of vaccination” is not included in Table 2, as none of the respondents marked this response.

**Table 3 vaccines-10-01206-t003:** Approach to vaccination implementation in own child vs. demographic categories (N = 403).

	I Vaccinate Only with Mandatory Vaccinations	I Vaccinate with Mandatory Vaccinations and Selected Recommended Vaccinations	I Vaccinate with Mandatory Vaccinations and All Recommended Vaccinations	I Do Not Vaccinate My Child at All	*p* Value
	N (%)				
Gender					
Female	11 (3.4)	154 (47.2)	161 (49.4)	0 (0.0)	*p* = 0.132
Male	1 (1.3)	40 (51.9)	35 (45.5)	1 (1.3)	
Age					
<40	3 (7.7)	19 (48.7)	17 (43.6)	0 (0.0)	*p* = 0.021
41–50	1 (1.5)	34 (51.5)	31 (47.0)	0 (0.0)	
51–60	3 (2.0)	77 (48.7)	77 (48.7)	1 (0.6)	
61<	4 (3.7)	47 (44.0)	56 (52.3)	0 (0.0)	
I do not want to specify	0 (0.0)	17 (51.5)	16 (48.5)	0 (0.0)	
Seniority in the profession					
Up to 5 years	0 (0.0)	2 (100.0)	0 (0.0)	0 (0.0)	*p* = 0.475
More than 5 to 10 years	0 (0.0)	10 (43.5)	13 (56.5)	0 (0.0)	
More than 10 to 15 years	2 (10.5)	11 (57.9)	6 (31.6)	0 (0.0)	
Over 15 years	10 (2.8)	171 (47.6)	177 (49.3)	1 (0.3)	
Self-assessment of knowledge about vaccines					
Very bad	0 (0.0)	0 (0.0)	0 (0.0)	0 (0.0)	*p* = 0.383
Bad	0 (0.0)	1 (50.0)	1 (50.0)	0 (0.0)	
Average	1 (5.6)	12 (66.7)	5 (27.8)	0 (0.0)	
Good	9 (4.3)	107 (50.7)	94 (44.5)	1 (0.5)	
Very good	2 (1.2)	71 (42.3)	95 (56.5)	0 (0.0)	
I do not know/hard to say	0 (0.0)	3 (75.0)	1 (25.0)	0 (0.0)	

**Table 4 vaccines-10-01206-t004:** Influenza vaccination by paediatricians (N = 403).

	Regularly Every Year or Almost Every Year	Irregularly from Time to Time	Never Get Vaccinated	*p* Value
	N (%)			
Gender				
Female	203 (62.3)	43 (13.2)	80 (24.5)	*p* = 0.973
Male	49 (63.6)	10 (13.0)	18 (23.4)	
Age				
<40	21 (53.9)	8 (20.5)	10 (25.6)	*p* = 0.169
41–50	37 (56.0)	10 (15.2)	19 (28.8)	
51–60	104 (56.8)	19 (12.0)	35 (22.2)	
61<	72 (66.3)	12 (11.2)	23 (21.5)	
I do not want to specify	18 (54.5)	4 (12.1)	11 (33.3)	
Seniority in the profession				
Up to 5 years	2 (100.0)	0 (0.0)	0 (0.0)	*p* = 0.681
More than 5 to 10 years	11 (47.8)	5 (21.7)	7 (30.4)	
More than 10 to 15 years	13 (68.4)	2 (10.5)	4 (21.1)	
Over 15 years	226 (63.0)	46 (12.8)	87 (24.2)	
Self-assessment of knowledge about vaccines				
Very bad	0 (0.0)	0 (0.0)	0 (0.0)	*p* = 0.213
Bad	1 (50.0)	1 (50.0)	0 (0.0)	
Average	13 (72.2)	0 (0.0)	5 (27.8)	
Good	121 (57.3)	32 (15.2)	58 (27.5)	
Very good	115 (68.5)	19 (11.3)	34 (20.2)	
I do not know/hard to say	2 (50.0)	1 (25.0)	1 (25.0)	

**Table 5 vaccines-10-01206-t005:** Means, Standard Deviations, and Correlations among variables.

Variable	Mean	SD	1.	2.	3.
1. Behavioural attitude	2.45	0.568	-		
2. Cognitive attitude	3.86	0.392	0.285 *	-	
3. Influenza self-vaccination	1.38	0.851	0.371 *	0.306 *	-

Note. * *p* < 0.001.

**Table 6 vaccines-10-01206-t006:** Total, direct, and indirect links of cognitive attitude and influenza self-vaccination of paediatricians through behavioural attitude.

				Bootstrap 95% Confidence Interval (CI)	
Effect (β)	SE	T	*p*	LLCI	ULCI
Total effect (βyx): cognitive attitude (X) on influenza self-vaccination (Y)					
0.619 F_p_ = 35.410 R^2^ = 0.081	0.103	5.95	<0.001	0.415	0.822
Direct effect: cognitive attitude (X) on influenza self-vaccination (Y)					
0.432	0.108	3.076	<0.001	0.218	0.645
Indirect effect (βyx.m): cognitive attitude (X) on influenza self-vaccination (Y) through the behavioural attitude (M)					
0.187	0.046			0.095	0.279

Notes: Lower-level confidence interval (LLCI); upper-level confidence interval (ULCI); number of bootstrap samples for bias-corrected bootstrap confidence intervals: 5000; Level of confidence for all confidence intervals in output: 95%. N = 403.

## Data Availability

The data presented in this study are available on request form the corresponding author.

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
