# Peer review of "Behavioural and Cognitive Attitudes of Paediatricians towards Influenza Self-Vaccination—Partial Mediation Model"

_vaccines, 2022, doi:10.3390/vaccines10081206_

Round 1
Reviewer 1 Report
1. Why did the authors select only influenza vaccination/immunization? Please justify and discuss.
2. Why only pediatricians? Please justify and discuss.
3. What will be the possible implications of this study on diseases other than influenza? Please discuss.
4. Line 25. What is Triandis theory? Readers may not be familiar with this, please rewrite it in an explanatory way.
5. What is the author's focus? Self-immunization or self-vaccination. Are these two different terms? How can one do self-vaccination? What exactly authors meant by this. Please justify and discuss.
6. Line150. Section 3.3 is missing. Please complete and update.
7. There are no references included in the whole material and methods sections. Please include the appropriate references.
8. The authors are talking about behavior while the ratio of males and females in the study is not close to equal? Is that not affecting the study conclusions? Please justify and discuss.
9. There are several limitations to this study. Please include a separate section about the limitations of the current study.
10. If possible authors should translate the results into some vein diagrams or graphs to make the results clear and make the study easy to understand among readers.
11. Authors should include a flow or representative diagram showing what authors did and what is the outcome. This is important and this will improve the readability of the study.
Author Response
Thank you very much for your careful reading of the article and your very important comments, which will undoubtedly affect the quality of the material presented and the readers' perception of the article.
Below we refer to the sent comments with additions and clarifications.
- Why did the authors select only influenza vaccination/immunization? Please justify and discuss.
We write about it in the Introduction, lines: 64-78
- Why only pediatricians? Please justify and discuss.
We write about it in the Introduction, lines 39-63
- What will be the possible implications of this study on diseases other than influenza? Please discuss.
We write about it in Conclusions and Practical Implications, lines: 612-629
- Line 25. What is Triandis theory? Readers may not be familiar with this, please rewrite it in an explanatory way.
Due to the limited number of words in the Abstract, we explain Triandis' Theory in Chapter 2 Assumptions of Theory of Interpersonal Behaviour, lines: 95-107.
- What is the author's focus? Self-immunization or self-vaccination. Are these two different terms? How can one do self-vaccination? What exactly authors meant by this. Please justify and discuss.
Thank you very much for bringing this to our attention. Using both terms in one article may confuse the reader, and self-immunization may be read as natural immunity. We mean self-vaccination in this case, so we have unified the term throughout the article.
- Line150. Section 3.3 is missing. Please complete and update.
We have changed a bit the article's design and moved this part to the Discussion. The description addresses hypothesis H3 in lines: 525-533
- There are no references included in the whole material and methods sections. Please include the appropriate references.
References relating to the national vaccination program were missing, and we have supplemented this.
- The authors are talking about behavior while the ratio of males and females in the study is not close to equal? Is that not affecting the study conclusions? Please justify and discuss.
The ratio of men and women is not equal and reflects the population of pediatricians in Poland. Pediatricians in Poland are mostly women. This does not affect the results of the survey, as the sample was selected in a random-quota manner and properly weighted, and for this reason, the results of the survey are not affected.
- There are several limitations to this study. Please include a separate section about the limitations of the current study.
We created a new subsection for Limitations to make it more visible, lines: 555-610
- If possible authors should translate the results into some vein diagrams or graphs to make the results clear and make the study easy to understand among readers.
As the survey data are crossed in the results, we decided to stay with tables, as they seem more readable.
- Authors should include a flow or representative diagram showing what authors did and what is the outcome. This is important and this will improve the readability of the study.
This is well illustrated in Figure 1, lines: 497-499
Reviewer 2 Report
In this article, the authors analysed the attitudinal components that influence self-vaccination among paediatricians within the immunization program, showing that pro-vaccination behaviours are not only influenced by cognitive factors but also the behavioural components of attitudes contribute.
Findings provide important evidence representing a first study of Polish paediatricians showing relationship attitudes and behaviour related to self-immunization.
The introduction provide sufficient background, including relative references.
The research design results appropriate.
The discussion is not enough consistent with the evidence and
arguments presented, requiring more correlations to previous similar studies.
The references result appropriate.
Author Response
The discussion is not enough consistent with the evidence and
arguments presented, requiring more correlations to previous similar studies.
Thank you very much for this comment. It prompted us to make changes in the design of the article so that we could contrast our results with the work of other authors on this topic.
Round 2
Reviewer 1 Report
The authors successfully responded to the reviewer's comments and updated the manuscript as well.